# Bone Marrow-Derived Mesenchymal Stromal Cells: A Novel Target to Optimize Hematopoietic Stem Cell Transplantation Protocols in Hematological Malignancies and Rare Genetic Disorders

**DOI:** 10.3390/jcm9010002

**Published:** 2019-12-18

**Authors:** Stefania Crippa, Ludovica Santi, Roberto Bosotti, Giulia Porro, Maria Ester Bernardo

**Affiliations:** 1San Raffaele-Telethon Institute for Gene Therapy (SR-TIGET), San Raffaele Scientific Institute, 20132 Milan, Italy; crippa.stefania@hsr.it (S.C.); santi.ludovica@hsr.it (L.S.); bosotti.roberto@hsr.it (R.B.); porro.giulia@hsr.it (G.P.); 2Pediatric Immunohematology and Bone Marrow Transplantation Unit, San Raffaele Scientific Institute, 20132 Milan, Italy

**Keywords:** mesenchymal stromal cells, bone marrow niche, hematopoietic stem and progenitor cells, hematopoietic stem cell transplantation, ex-vivo gene therapy

## Abstract

Mesenchymal stromal cells (MSCs) are crucial elements in the bone marrow (BM) niche where they provide physical support and secrete soluble factors to control and maintain hematopoietic stem progenitor cells (HSPCs). Given their role in the BM niche and HSPC support, MSCs have been employed in the clinical setting to expand ex-vivo HSPCs, as well as to facilitate HSPC engraftment in vivo. Specific alterations in the mesenchymal compartment have been described in hematological malignancies, as well as in rare genetic disorders, diseases that are amenable to allogeneic hematopoietic stem cell transplantation (HSCT), and ex-vivo HSPC-gene therapy (HSC-GT). Dissecting the in vivo function of human MSCs and studying their biological and functional properties in these diseases is a critical requirement to optimize transplantation outcomes. In this review, the role of MSCs in the orchestration of the BM niche will be revised, and alterations in the mesenchymal compartment in specific disorders will be discussed, focusing on the need to correct and restore a proper microenvironment to ameliorate transplantation procedures, and more in general disease outcomes.

## 1. Introduction

Mesenchymal stromal cells (MSCs) are a rare population of non-hematopoietic multipotent cells resident in the bone marrow (BM), which offer physical support and regulate hematopoietic stem/progenitor cell (HSPC) homeostasis. MSCs were first isolated from the BM [1,2], thanks to their ability to adhere to plastic and generate colony-forming unit fibroblasts (CFU-Fs) in vitro. MSCs can be easily expanded for several passages as fibroblast-like cells. In vitro, they are positive for the expression of specific surface markers, classification determinant (CD)105, CD90, and CD73, whereas they do not express hematopoietic (CD34, CD45) and endothelial markers (CD31). They express human leukocyte antigen (HLA) class I but they are negative for HLA class II. MSCs can differentiate into skeletal, connective, and adipose tissue when exposed to proper conditions [3].

In the human BM, MSCs are localized around the blood vessels, where they offer physical support to HSPCs and differentiate into osteoprogenitors to guarantee a functional remodeling of the BM niche. Importantly, BM-MSCs control HSPC homeostasis by direct contact and in a paracrine manner through the secretion of soluble factors [4,5,6]. The concept that MSCs play a fundamental role in the regulation of hematopoiesis is supported by data showing the co-localization of MSCs with sites of hematopoiesis, starting from embryonic developmental stages [7]. The understanding of MSC’s role in the BM niche has been limited for a long time due to the difficulty of identifying specific markers to localize and prospectively isolate MSCs in vivo. The lack of consensus on surface markers has generated contradictory results on independent subpopulations of MSCs [8,9,10,11,12,13,14,15]. However, recent studies have clarified the identity of MSC subsets which are mainly involved in the control of HSPC homeostasis. Sacchetti et al. first reported that MSCs positive for the CD146 marker reside in the sinusoidal wall, are enriched for colony forming unit-fibroblast (CFU-F) activity, and can generate a BM niche supporting hematopoietic activity when transplanted heterotopically in immunodeficient mice. CD146^+^ cells express HSPC regulatory genes such as Angiogenin-1 and C-X-C motif chemokine 12 (CXCL12) [11]. Later, CD271 has been used to identify MSCs localized in the trabecular region of human BM. CD271^+^ MSCs show an enhanced clonogenic and differentiation capacity and express higher levels of extracellular matrix and cell adhesion genes compared to bulk MSCs [16,17,18]. These data suggest that different subtypes of MSCs interact with HSPCs in specific perivascular regions. CD271^+^ and CD271^+^/CD146^-/low^ MSC have been reported to be bone-lining cells associated with long term (LT)-HSPC in low oxygen areas, whereas CD146^+^ and CD271^+^/CD146^+^ are located around BM sinusoids in association with proliferating HSPCs [12] (Figure 1). Increasing evidence supports the hypothesis that MSCs represent a subpopulation of pericytes associated with the vessels of multiple human tissues. For this reason, MSCs/MSC-like cells have been isolated from several adult tissues, including adipose tissue, heart, skin, Wharton’s jelly, dental pulp [19,20,21]. Despite the broad anatomical distribution, the majority of available data on MSC functionality have been obtained with ex-vivo expanded MSCs due to their low frequency. In human BM, MSCs represent 0.001–0.01% of mononuclear cells, thus requiring extensive ex-vivo manipulation for their functional characterization and clinical application [13]. Published data indicate that MSCs may become heterogeneous and acquire different properties upon plastic adherence and culture media exposure [22,23,24]. It has been shown that MSC cultures undergo clonal selection during the expansion phase, and selected clones possess different capabilities [25]. Moreover, MSC function is the result of coordinated interactions with the other BM niche components and may operate differently in vitro. Abbuehl et al. recently demonstrated that freshly-isolated murine BM-MSCs, but not ex-vivo expanded, are capable of engrafting long-term and to repair stromal niche damage after irradiation, translating into a significantly better HSPC engraftment after co-transplantation with HSPC intra-bone [26]. Genome-wide analysis has revealed a distinct transcriptional profile of human primary MSCs and corresponding in vitro counterpart, highlighting an enhanced hematopoietic supportive function in primary MSCs [22]. For this reason, the manipulation of culture conditions, including cytokines, glucose concentration, oxygen tension, culture as mesenspheres, have been proposed as a strategy to maintain MSC native properties [27,28,29], including their capacity to support HSPCs. More recently, the use of a cocktail of transcription factors has been shown to reprogram murine ex vivo expanded MSCs to a more primitive state [30].

MSCs have been employed as tools for tissue engineering and regenerative medicine [31,32,33], as well as a cell-therapy approach to counteract inflammation in immune-mediated and inflammatory diseases thanks to their capacity to sense the inflammatory state and suppressing immune responses [34,35,36,37,38,39]. Moreover, several pre-clinical and clinical studies have highlighted the role of ex-vivo cultured MSCs in promoting hematopoietic recovery and reducing the risk of graft failure [40,41]. MSCs have also been successfully employed to stimulate ex-vivo expansion and maintenance of CD34^+^ HSPCs [42,43].

Given their role in the BM niche and in HSPC support, dissecting the in vivo function of human MSCs and studying their properties in pathological conditions is a fundamental requirement to optimize the outcome of HSPC transplantation (HSCT) and ex-vivo HSPC-gene therapy (HSC-GT). As indicated in the editorial of the Special Issue “Mesenchymal Stromal Cell-Based Therapy: New Perspectives and Challenges” [44], also in the present review novel findings on the role of MSCs in the BM niche will be reported. In particular, alterations in the mesenchymal compartment in hematological malignancies, as well as in genetic disorders treated with HSCT or HSC-GT, will be discussed, focusing on the need to correct possible defects and preserve MSC functionality before transplantation.

## 2. MSCs as Key Elements of the Bone Marrow Niche

MSCs play a pivotal role in the control of hematopoiesis in the BM niche since the embryonal stage. MSCs have been identified in the aorta-gonad-mesonephros (AGM) region at E11, following the hematopoietic system development [7].

These cells have been extensively studied in vivo in several animal models, elucidating important aspects of MSC biological characteristics and function in the BM niche. In murine BM, MSCs occupy specific anatomic positions defined as endosteal and vascular niche [5,45]. In particular, MSCs localize in the endosteal niche, lining the bone surface and physically interacting with both osteoblasts and HSPCs [6,46,47]. Endosteal MSCs represent a source of osteoprogenitors and indirectly contribute to osteogenesis by secreting several growth factors and cytokines [48]. It has been shown that MSCs enhance bone regeneration [49] and mediate the capacity of parathyroid hormone to expand the osteoblast population, which, in turn, facilitates the expansion of primitive HSPCs through the secretion of several hematopoietic growth factors (granulocyte colony-forming factor, angiopoietin, interleukin 6 CXCL12) and the activation of Notch signaling [50,51].

In the vascular niche, MSCs are associated with blood vessels in a perivascular position, regulating HSPCs homeostasis by direct interaction with HSPCs or in a paracrine manner through the secretion of hematopoietic supportive factors [52,53]. Murine Nestin^+^ MSCs have been first described as perivascular MSCs closely in contact with HSPCs. These cells are associated with blood vessels of the central marrow and are present at a lower frequency, in the endosteum. They are also in contact with adrenergic nerve fibers of the sympathetic nervous system, known to play an important role in regulating HSPC mobilization. Importantly, they express several HSPC maintenance factors, such as CXCL12, stem cell factor (SCF), angiopoietin-1, IL-7, vascular cell adhesion molecule 1 (VCAM1) [6,54].

Interestingly, Nestin^+^ MSCs show several similarities to recently identified CXCL12-abundant reticular cells (CAR) [55]. CAR cells are also associated with sinusoidal endothelium; they have a morphology similar to vascular pericytes and sustain the maintenance of primitive HSPCs [56,57]. The reduction of the HSPC pool upon short-term ablation of CAR cells highlights the fundamental role of CAR cells in sustaining primitive HSPCs in a mouse model. In addition, the absence of CAR cells causes the upregulation of PU.1 in HSPCs and their commitment towards the myeloid lineage [58]. However, Nestin+ cells are considered a more primitive MSC population, since they are less abundant than CAR cells, show a higher clonogenic capacity, self-renewal activity, and differentiation potential [59]. These results sustain the concept of the BM stroma composed of different MSC subpopulations, localized in specific interconnected areas, which allows the crosstalk among different types of cells and molecular signaling.

Similarly, in the human BM niche, MSCs are localized in the endosteum and around the sinusoidal vessels [12]. Despite the study of the human BM niche, it is challenged by their difficult accessibility; different subpopulations of MSCs have been identified in vivo based on the expression of CD146 (MCAM), CD271 (Nerve Growth Factor Receptor), Stro-1, and stage-specific embryonic antigen-4 (SSEA-4) markers. CD271 is expressed by bone lining MSCs proximal to the surface of trabecular bone. CD271^+^ MSCs show a robust clonogenic capacity with an increased proliferation potential and ability to differentiate into mesodermal tissues compared to the CD271 negative MSCs [16]. Transcriptional analysis of CD271^+^ MSCs sorted from the BM confirmed enhanced expression of extracellular matrix and cell adhesion genes and increased levels of early osteogenesis/chondrogenesis/adipogenesis genes [18]. CD271^+^ MSCs have been shown to have enhanced capability in promoting HSPC engraftment [17]. However, the majority of CD271^+^ MSCs do not co-express CD90 and CD73, two of the markers included in the minimal criteria used to define ex-vivo expanded MSCs, according to the International Society for Cellular Therapy position paper [3]. For this reason, CD271^+^ expression is not an adequate marker to prospectively isolate bonafide MSCs from BM samples, although it is not clear whether the expression of CD90, CD73, and CD105 represents an in vitro artifact. Preliminary data obtained in our laboratory demonstrate the existence of a rare population of cells positive for CD90, CD73, CD105 in the CD34 negative fraction of BM aspirates. Moreover, CD271 shows different expression levels among various MSC subsets, and it is not universally expressed in MSCs derived from different tissues, suggesting that this marker is not sufficient to prospectively isolate MSCs [8,60]. For this reason, low/negative expression of PDGFR-α has been combined with CD271 expression to identify the candidate primary MSCs within the CD45^−^CD271^+^ cell population [13]. CD146^+^ MSCs are localized around the sinusoidal vessels similar to pericytes, show high clonogenic capacity, and are able to re-establish the hematopoietic microenvironment when transplanted into xenograft models [11,61]. CD146 expression also defines MSCs with higher multipotency [62]. Importantly, CD146 expression differentiates between perivascular and endosteal localization of MSCs. Evidence to date suggests that CD271^+^ and CD271^+^/CD146^-/low^ MSCs are bone-lining cells associated with long-term (LT)-HSPCs in low oxygen areas, whereas CD146^+^ and CD271^+^/CD146^+^ are located around BM sinusoids in association with proliferating HSPCs. In both regions, HSPCs are located close to MSCs [12], which control their fate by secreting specific factors or by regulating the activity of the other niche components (Figure 1).

The use of Stro-1 and SSEA-4 as MSC markers is more debated. Stro-1 has been used in combination with negative selection for glycophorin-A to isolate highly clonogenic and multipotent MSCs from the BM [63]. However, in vivo studies demonstrated that Stro-1 negative MSCs support HSPCs engraftment in NOD/SCID mice [64]. SSEA-4 identifies a subpopulation of MSCs with high proliferation capacity, capable of differentiating into osteoblasts [65]. Nonetheless, other studies have demonstrated that SSEA-4 expression is an in vitro artifact due to serum exposure [66].

In light of these data, a deeper biological characterization of human BM-MSC subpopulations is required to better clarify their localization within the niche, the expression of specific surface markers for their prospective isolation, and their interaction with HSPCs. In this sense, the development of humanized niche models represents a powerful tool to dissect the hematopoietic supportive function of different MSC subpopulations in vivo within a microenvironment mimicking the human situation [67,68].

## 3. MSCs in the Clinical Use

Thanks to their biological and functional properties, MSCs have emerged as a new therapeutic strategy for a wide range of clinical conditions. In particular, MSCs provide support to HSPCs in the BM niche and display potent inflammatory sensing capacities and immunomodulatory functions. Indeed, MSCs have been shown to interact and regulate the activity of innate and adaptive immune cells, despite several differences existing according to MSCs’ tissue sources [69]. In particular, MSCs are able to inhibit B-, T-, and natural killer (NK)-cell proliferation by direct interaction and through the release of soluble molecules, such as transforming growth factor (TGF)-β1 [70], indoleamine 2,3-dioxygenase (IDO) [71], prostaglandin E2 (PGE2) [72], which induces immune cells to arrest in G0, preventing their expansion. Besides, MSCs control T-cell activation inhibiting the production and secretion of inflammatory cytokines, and prevent the activation and maturation of dendritic cells [73]. Based on these properties, MSCs are considered an attractive “tool” to manage immune-mediated disorders, such as Graft-versus-Host-Disease (GvHD), a complication arising in patients undergoing allogeneic HSCT and associated with a high mortality rate. In 2004, Le Blanc first demonstrated in a seminal case report that the intravenous infusion of ex-vivo expanded BM-derived MSCs can effectively control manifestations of steroid-resistant, acute GvHD, in an allogeneic context. In the following Phase II trial, of the 55 patients treated with third-party donor MSCs for acute GvHD following allogeneic HSCT, 30 had a complete remission, and nine showed partial improvement. Indeed, these data demonstrate that MSCs might be a safe and effective treatment for patients suffering from acute GvHD who do not respond to standard immunosuppressive therapies [74]. Hereafter, other Phase II clinical trials have confirmed these results in terms of safety and efficacy of third-party donor MSC administration in the GvHD setting, suggesting that the timing of MSC infusions is critical to increasing their effectiveness and that the inflammatory state in the patients plays a fundamental role in the immunoregulatory activity of MSCs [75]. For these reasons, a precocious infusion after GvHD development and multiple administrations have been suggested as the best options for successful control of the disease [76]. Concerning chronic GvHD, the impact of MSC infusion is still debated, although combining MSCs and immunosuppressive therapies may be a safe and effective therapeutic option [77].

In addition to immunoregulatory functions, MSCs are known to promote and support HSPC engraftment. In vitro MSCs enhance the expansion and maintenance of HSPCs [78], whereas they promote long-term engraftment when co-transplanted with HSCPs in different animal models [79,80,81]. The molecular mechanism underlying the supportive effect of MSCs on HSC engraftment is still controversial. However, the capacity of MSCs to reduce inflammation in the BM niche and support HSPCs by secreting soluble factors render MSCs an attractive tool to ameliorate transplantation outcomes [82]. Indeed, conditioning regimens based on radiotherapy and chemotherapy administered before HSCT may profoundly affect the BM microenvironment, thus delaying hematopoietic reconstitution. MSCs have been co-infused with HSPCs in Phase I/II clinical trials with the aim of facilitating hematopoietic engraftment and reducing the risk of graft rejection, demonstrating its safety and efficacy [41,82]. Taking advantage of their hematopoietic supportive function, MSCs have been exploited to enhance HSPC expansion in vitro before HSCT with umbilical cord blood (UCB)-derived cells resulting in faster neutrophil and platelet recovery [42,43]. The use of MSCs is intended to mimic the stromal compartment of the BM niche by providing supportive factors for HSPC expansion and maintenance. Although MSCs have been demonstrated to preserve and support HSPC proliferation been through co-culture systems [42,83], differences may exist according to the source of MSCs. For example, placenta-derived MSCs have been reported to be a better feeder than UCB-derived MSCs, as they could maintain HSPCs in a more primitive state [84].

MSCs have also exploited clinically in the field of Regenerative Medicine, considering their ability to differentiate into bone and chondrocytes, especially when combined with biomaterial scaffolds. Scaffold employment and stimulating factors, such as BMP-2, are generally used to promote successful osteoblast differentiation for the regeneration of skeletal tissues [85]. Use of MSCs to repair damaged cartilage and other tissues, such as tendons, ligaments, and intervertebral discs, represents a novel and promising therapeutic strategy, especially for those tissues with avascular nature [86,87]. MSCs have been also employed for the regeneration of the central nervous system (CNS), heart and liver, cornea, and trachea [88]. In these cases, MSCs are thought to promote regeneration by secreting paracrine factors, despite there is no clear explanation. However, recent studies suggest that the main driving force behind the therapeutic efficacy of MSCs is the paracrine factors secreted by these cells and propose the administration of MSC-conditioned media for regenerative medicine applications [89]. In particular, MSCs have been demonstrated to provide beneficial effects in the treatment of neurological diseases, such as Parkinson [90], ischemic stroke [91], and multiple sclerosis [92]. Also, MSCs have been tested to promote- myocardium and liver regeneration.

In conclusion, MSCs are emerging as a promising strategy for cellular-based therapies in the context of inflammatory and immune-mediated diseases thanks to their ability to modulate inflammatory responses and immune cells. In the field of HSCT, MSCs have been extensively studied for their capacity to sustain HSPC and facilitate their engraftment. These properties have been exploited both in vitro in expansion strategies and in vivo in co-infusion approaches with the final goal to optimize HSCT outcome. Finally, the ability to differentiate and release growth factors in a damaged microenvironment render MSCs a valid tool for tissue regeneration. However, further pre-clinical studies and clinical trials are required to better elucidate the molecular mechanisms responsible for the therapeutic efficacy of MSCs. Furthermore, it must be taken into account that manufacturing MSCs in vitro, before administration, may affect some of their biological characteristics. The culture conditions should be refined and optimized to preserve primary MSC biological and functional properties and to minimize culture-induced alterations [93,94,95].

## 4. MSCs in Hematological Malignancies

In recent years an active role of the BM niche is emerging in the pathogenesis of human hematological malignancies. In particular, alterations in the mesenchymal compartment have been described to support the expansion and survival of leukemic stem cells (LSCs). The abnormal activity of MSCs is mainly caused by tumorigenic signals, which render the BM stroma immunologically tolerant to tumor growth and instruct MSCs to sustain LSCs at the expense of normal hematopoiesis. Several studies observed a reduced proliferative capacity of MSCs isolated from BM samples of patients affected by different hematological malignancies. In particular, it has been reported that MSCs derived from acute myeloid leukemia (AML) patients have a reduced capacity to form CFU-Fs, with a failure of 25% in MSCs isolation, and a lower population doubling compared to healthy controls [96]. AML-MSCs display an abnormal adipogenic potential [97], together with impaired osteogenic capacity and a diminished capacity to support CD34^+^ HSPCs [98]. In line with these observations, a genome-wide analysis of AML-MSCs confirmed reduced expression of hematopoietic supportive genes, including KIT ligand (KITLG), thrombopoietin (THPO), and angiopoietin (ANGPT1), associated with a reduced proliferation capacity and a perivascular signature of leukemic MSCs [99]. This defect was associated with an altered methylation profile of AML-MSCs, which impaired the expression of several genes fundamental in the BM niche. For instance, reduced expression of KITLG and overexpression of jagged canonical Notch ligand 1 (JAG1) in AML-MSCs was reported to favor BM niche support to LSCs. In particular, NOTCH/JAG1 signaling resulted in playing a fundamental role in MSC-dependent control of tumor initiation, growth, and chemoresistance. JAG1 overexpression has been shown to induce AML in mice [100], whereas NOTCH1, JAG1, and the main Notch target gene HES1 are overexpressed in AML-MSCs thus promoting survival of tumor cells exposed to chemotherapeutic agents [101] (Figure 2). Several studies have demonstrated that the tumor BM milieu is sufficient to convert a healthy BM stroma into a tumor supporting microenvironment. For example, the exposure of healthy donor MSCs to AML-MSC conditioned medium is sufficient to decrease MSC proliferation and osteogenic differentiation [98]. Other studies reported an increased immunosuppressive/anti-inflammatory potential in AML-MSCs compared to controls. In particular, IL-10 secretion by AML-MSCs resulted in correlating directly with overall patient survival [102,103].

Similar to AML-MSCs, MSCs isolated from myelodysplastic (MDS) patients show a reduced proliferation rate [96]. Several pieces of evidence suggest a fundamental role of MSCs in the initiation of MDS in aged patients. Indeed, aged MSCs undergo replicative senescence and activate a specific senescence secretome in response to stress signals, including abnormal mitogenic signals, oxidative and genotoxic stress [104,105]. Among these secreted factors, several inflammatory cytokines sustain chronic inflammation, which could initiate and sustain cancer progression [106,107]. MDS-MSCs display in vitro several senescence features, including large, flat, and granular morphology, impaired proliferation, and increased β-galactosidase expression [108,109]. MSD-MSCs undergo premature cell-cycle arrest due to a significant upregulation of cyclin-dependent kinase inhibitor 2B (CDKN2B) compared to healthy counterparts, highlighting a possible role of MSCs in the control of tumor growth. Moreover, MDS-MSCs has been shown to support LSC expansion by different mechanisms specifically. These cells produce high levels of IL-6, which may modify HSPC biology. Indeed, from one side, IL-6 may induce HSPCs differentiation to the detriment of self-renewal, and, from the other side, it might increase LSC proliferation. Thus, MSCs, or better, dysfunctional MSCs, may affect cancer microenvironment [110]. Similarly, MDS cells proliferate to a greater extent on MDS-MSCs compared to normal MSCs. This is due to the downregulation of metalloproteinase 1 (MMP1), which renders MDS-MSCs unable to induce apoptosis in cancer cells [111]. Moreover, it has been shown that the expression of several hematopoietic supportive factors was reduced in MDS-MSCs, including KITLG and Angiopoietin-1. This results in a diminished cell cycle activity of HSPCs cultured on MDS-MSCs, highlighting that impaired stromal support contributes to ineffective hematopoiesis [98,112,113]. Moreover, several studies demonstrated an altered immunomodulatory function of MDS-MSCs, affecting tumor immune surveillance [114]. Evidence to date suggests that MSCs from neoplastic niche are educated to polarize macrophage toward an M2, anti-inflammatory, phenotype and they are responsible for increase Treg through the secretion of indoleamine 2,3-dioxygenase (IDO). In addition, MDS-MSCs have been shown to inhibit dendritic cell maturation and alter their functions, including endocytosis, IL-12 secretion, their ability to inhibit T cell proliferation. [115,116,117]. In this light, BM stroma contributes to the creation of a protective and immune-tolerant microenvironment capable of supporting the survival of leukemic cells and affect the response to therapies.

MSCs have also been isolated and characterized from BM samples of patients affected by acute lymphoblastic leukemia (ALL). ALL-MSCs isolated from children at different times in the course of the disease show reduced proliferation, increased adipogenic capacity, and impaired supportive function when co-cultured with HSPCs [118]. ALL-MSCs show a significantly higher level of pro-inflammatory cytokines, including IL-8 and CXCL2 [119], and express α-smooth muscle actin, linked with cancer-associated fibroblasts that contribute to the acquisition of invasive phenotypes (CAFs) [120].

The dysregulation of BM stroma is mainly the result of extensive crosstalk between cancer cells and MSCs. However, a novel concept highlighting the primary role of MSCs in tumor initiation is emerging. In particular, it has been shown that primary stroma alterations can induce a malignant transformation of the hematopoietic compartment in different mouse models. In the first model, the deletion of Dicer1 in mouse osteoprogenitors impairs osteogenic differentiation and causes ineffective hematopoiesis with myelodysplasia [121]. Malignant cells acquire several genetic abnormalities while having intact Dicer1. Importantly, myelodysplasia is environmentally induced. When wild-type BM is transplanted into mutant mice or control mice, mutant recipients develop signs of myelodysplasia. Similarly, Dicer null osteoprogenitors induce abnormal modification of HSPCs in vitro. Dicer1 deletion is associated with reduced expression of the ribosome maturation factor Shwachman-Bodian-Diamond Syndrom (SBDS), encoded by the gene mutated in Schwachman–Bodian–Diamond syndrome, a human BM failure, and leukemia pre-disposition condition. Importantly, decreased expression of Dicer1 was detected in MSC-derived osteoprogenitors from myelodysplastic syndrome (MDS) patients, along with a reduction of the SBDS gene [122]. Similarly, conditional loss of nuclear factor kappa B (NFkB) inhibitor in stromal cells causes upregulation of JaG1/Notch signaling in HSPCs, resulting in a disorder similar to chronic myelomonocytic leukemia (CMML). On the contrary, constitutive activation of the NFkB pathway in myeloid cells does not recapitulate in a cell-autonomous manner the leukemia phenotype, clearly indicating that the malignant transformation of hematopoietic cells is initiated by BM stroma [123]. Increased expression of Notch1 and Jagged1 has been observed in cell lines from patients with AML [124]. Moreover, the deletion of the Retinoic Acid Receptor γ (RARγ) in mice resulted in a chronic myeloproliferative disorder. Transplant studies revealed that RARγ-hematopoietic cells functioned normally when transplanted into normal mice. However, transplantation of normal hematopoietic cells into the RARγ-microenvironment resulted in a myeloproliferative disorder in the transplanted cells, revealing the capability of the microenvironment to be the only cause of hematopoietic disorders [125]. Finally, the inactivation of the retinoblastoma (RB) gene in the hematopoietic system induces myeloproliferation, which is due to the mobilization and differentiation of HSPCs from the BM. HSPC homeostasis is preserved when mutated HSPCs are transplanted into wild-type recipients, highlighting an RB-dependent interaction between BM stroma and HSPCs as causative of malignant transformation. Importantly, a mutation in the retinoblastoma (RB) pathway has been described in a vast majority of multiple myeloma (MM) cases [126], which is the best-studied example demonstrating that the interaction of hematopoietic cells, in this case, B cells, and the BM microenvironment is a major contributor to disease [127,128]. In particular, myeloma cells directly interact with BM stroma or extracellular matrix through various adhesion molecules that lead to the activation of proliferation and anti-apoptotic pathways. On the other side, these interactions trigger the stromal compartment to release a variety of cytokines that support tumor cell growth.

In conclusion, MSCs represent a key component of the BM niche regulating HSPC homeostasis. Several works have described specific alterations of MSC functional characteristics in hematological malignancies that in the majority of the cases, are induced by tumor cells. Primary alterations in the BM stroma have also been described and demonstrated to be sufficient to initiate malignant transformation.

## 5. Targeting BM Stroma: A Novel Therapeutic Approach to Treat Hematological Malignancies

Considering the emerging active role of the BM microenvironment in the pathogenesis of hematological malignancies, the possibility to target the BM niche to contribute to the eradication of the disease has been evaluated in several clinical trials as an adjuvant treatment (Figure 3). One of the best representative cases is the use of Denosumab, a monoclonal antibody inhibiting receptor activator of nuclear factor-kappa-Β ligand (RANKL) in multiple myeloma (MM) patients. RANKL expression is increased in MSCs of patients affected by MM, altering the normal balance of bone formation/reabsorption that is the cause of osteolytic lesions, bone pain, and related pathological fractures [129,130]. Several clinical trials demonstrated that Denosumab is capable of ameliorating bone disease in MM [131,132] and was associated with prolonged survival of treated patients, confirming an active role of BM stroma in the progression of the pathology [133]. The use of Dickkopf-related protein 1 (DKK-1) antagonist has been studied for their ability to restore osteoprogenitor function and prevent osteolytic bone lesions [134]. DKK1- plays an important role in MM-induced osteolysis by inhibiting osteoblast differentiation [135]. Besides, a key role of IGF1 in skeletal lesions formation has been described in MM patients [136]. The administration of small insulin growth factor-1 receptor (IGF-1R) inhibitors blocks the interaction of IGF1 released by BM stroma and tumor cells, causing a dose-dependent inhibition of cell proliferation and induction of cell death (Figure 3). Importantly, the altered balance of bone formation/bone resorption is not only the cause of bone lesions but also induces the release of proliferation/pro-survival factors from the BM stroma, which favor malignant cells at the dispense of HSPCs, favoring cancer progression [137]. For instance, the release of tumor necrosis factor-α (TNF-α) induces a change in the adhesion molecule profile of MM cells, which become more adherent to the BM niche displacing normal HSPCs [138]. This mechanism guarantees a higher level of protection from differentiation signaling and chemotherapeutic agents [139,140]. In this view, the VLA4-CD44 axes have been shown to increase the adhesion of cancer cells to MSCs and facilitate drug efflux [141]. Similarly, overexpression of N-cadherin mediates the adhesion of malignant cells to MSCs by increasing the number of N-cadherin-β-catenin complexes that promote the activation of Wnt signaling, mediating the resistance against tyrosine kinase [142]. An active form of TGF-β is released upon bone remodeling, increasing the production of IL-6 in MSCs and tumor cells, which plays an essential role in promoting cancer progression [143,144]. The upregulation of C-X-C chemokine receptor type 4 (CXCR4) is another strategy adopted by cancer cells to find protection within the BM niche. Indeed, CXCR4-CXCL12 interaction is one of the best-studied players in the cross-talk between cancer cells and BM-MSCs. CXCR4 has been shown to be overexpressed in several types of hematological malignancies, conferring an increased capacity of LSCs to seed in the BM niche where the microenvironmental conditions are more conducive to cell proliferation and viability [145,146,147].

Therefore, there are several levels of intervention to modulate the interaction of tumor cells with the BM stroma and reach a therapeutic effect. For instance, blocking CXCL12 binding to CXCR4 with the use of plerixafor (AMD3100) renders MM cells more susceptible to chemotherapeutic drugs through their mobilization into the circulation [148]. Similarly, several monoclonal antibodies against cell adhesion molecules (such as integrins) have been successfully evaluated pre-clinically to mobilize malignant cells from the BM [149]. Several drugs have been developed to neutralize the effect of IL-6 [150]. Conventional agents, such as INF-γ and all-trans retinoic acid (ATRA), has been shown to inhibit MM cell growth [151,152]. More recently, histone deacetylase (HDAC) inhibitors have been used to suppress the stromal production of IL-6 in co-culture models of MM cells and BM-derived MSCs [153,154].

Overall, these results support the central role of the BM stroma in the pathogenesis and progression of cancer. Considering that HSCT is curative in many of the disorders under discussion, a proper correction of tumor BM stroma is emerging as a strategy to ameliorate transplantation outcomes. Indeed, stromal cells remain of host origin after transplantation [155]. There have been reports of patients who are unable to achieve engraftment despite numerous attempts at HSCT [156], as well as cases in which leukemia arises after transplantation in donor cells [157,158,159] and one may speculate that these patients represent groups that do indeed have an underlying stromal defect which may benefit from targeted correction.

## 6. MSCs in Rare Genetic Diseases

The concept of a properly functional BM niche as a key requirement for the outcome of HSPC transplantation could be extended to all those diseases for which this procedure is indicated as a curative option, including a wide variety of rare genetic diseases, ranging from defects in the immune system, HSPC functionality, or metabolic diseases. In this sense, profound knowledge of BM stroma biology is fundamental for the optimization of transplantation strategies, especially in the context of HSC-GT, where the relationship between HSPCs and the stroma may influence harvest and behavior in culture, as well as HSC engraftment kinetics after gene modification. Indeed, the functionality of HSPCs could be affected by the diseased microenvironment, and, on the other side, the diseased niche could have a reduced capacity to sustain the engraftment of gene-corrected HSPCs [160,161,162,163].

In support of this hypothesis, different studies have highlighted specific alterations in the MSCs isolated from the BM samples of patients affected by genetic disorders that are treated with (hematopoietic stem progenitor cell transplantation (HSPCT) or HSC-GT. These results evidence the need for novel strategies to correct the BM niche before or during transplantation and the development of biological conditioning to avoid further damage of the BM stroma to improve transplantation protocols, by sparing their supporting niche [164,165]. In particular, MSCs have been isolated and characterized from pediatric patients affected by primary immune-deficiencies (PIDs) and undergoing HSCT or HSC-GT. Despite the fact that primary immunodeficiencies-mesenchymal stromal cells (PID-MSCs) do not display any alterations in terms of proliferation, differentiation, expression of MSC surface markers, the immunomodulatory capacity of PID-MSCs **is** altered compared to age-matched controls [166,167]. In particular, MSCs isolated from ADA-SCID, Wiskott–Aldrich Syndrome, Chronic Granulomatous Disease, and other SCID patients showed decreased inhibitory effect on T-cell proliferation. This is accompanied by a dysregulated production of pro- and anti-inflammatory cytokines. Alterations have also been observed in B-cell inhibition and maturation in the presence of PID-MSCs. On the other hand, priming of PID-MSCs with Toll-like receptor (TLR)3 and TLR4 evidence defects in the production of immunoregulatory molecules [167]. These data highlight defects in the immunoregulatory properties of PID-MSCs, which may be exhausted due to frequent, ongoing infections and extensive inflammation characteristic of PID patients (Figure 4). Importantly, from a clinical view, the altered secretory profile of PID-MSCs could impair their capacity to support gene-corrected HSPCs after gene therapy (GT). Indeed, a decreased functional activity and tendency to differentiation and exhaustion of HSPCs have been demonstrated in CGD patients [168].

MSCs have also been isolated from the BM aspirates of Mucopolysaccharidosis type I Hurler (MPSIH) patients, where the intracellular accumulation of glycosaminoglycans (GAGs) causes multiorgan dysfunction, including skeletal defects [169]. Considering the fundamental role of bone remodeling in support of HSPC homeostasis [5], the capability of MPSIH-MSCs to differentiate into functional osteoblasts has been evaluated in vitro and in vivo. Although MPSIH-MSCs can differentiate into osteoblasts, an unbalanced bone remodeling status has been noticed; indeed, an upregulation of RANKL/RANK/OPG (osteoprotegerin) has been found in patient-derived MSCs compared to controls, indicating increased ability to support osteoclastogenesis [170] (Figure 5). An extensive characterization of HSPCs in MPSIH patients is missing and may highlight possible defects in HSPC homeostasis, taking into consideration that osteoclasts play a fundamental role in HSPC mobilization and other functions [171,172,173]. An important aspect to be considered is that the extensive ex vivo culture of MPSIH-MSCs may change their functional characteristics, reducing the extent of the functional defects [30]. The culture of MSCs in conditions of glycosaminoglycans (GAGs) overload may represent a valid strategy to reproduce in vitro the pathological conditions of the MPSIH BM niche. Following this hypothesis, MSCs isolated from BM samples of transfusion-dependent β-thalassemia (BT) patients **have** been exposed in vitro to iron to study their antioxidant response in a condition mimicking the iron overload state of the BM niche. BT-MSCs show an impaired clonogenic capacity, reduced proliferation rate, and altered differentiation potential. A condition of iron overload has been identified in the BM niche of BT patients causing a significant upregulation of ROS level in BT-MSCs. The exposure of BT-MSCs to increasing doses of iron revealed an inappropriate antioxidant response, which is responsible for the pauperization of the most primitive MSC fraction [174]. In addition, β-thalassemia-mesenchymal stromal cells (BT-MSCs) express lower level of hematopoietic supportive factors compared to controls, that negatively affect their ability to attract HSPCs in vitro, to sustain HSPC expansion and primitive phenotype in 2D co-culture model, to favor HSPC engraftment and immunological reconstitution in xenogenic transplant models, and to form a proper BM niche in vivo [174] (Figure 6). These results highlight a profound defect in the BM niche of BT patients, which may explain the increased risk of graft rejection and mixed chimerism observed after HSTC [175,176] and prove the need to treat the BM niche with the aim of reducing oxidative stress, thus potentially ameliorating transplantation outcome.

In light of these studies, the use of MSCs in co-transplantation strategies in the clinical setting to facilitate the engraftment of HSPCs [41,177] and to promote the rescue of the resident stromal compartment may be evaluated in some pathologies. Indeed, considering the disease-specific defects observed in patient-derived MSCs, co-administration of third-party, healthy donor-derived MSCs is to be preferred in co-transplantation settings [178]. In conclusion, recent data show specific alterations in the mesenchymal compartment of patients affected by rare genetic disorders and evidence the need to investigate the functional properties of diseased MSCs, to restore a proper microenvironment supporting HSPC engraftment and long-term hematopoiesis, with the final goal to improve the efficacy of HSC transplantation.

## 7. Conclusions

HSCT is an available curative option for several hematological malignant and nonmalignant disorders, including genetic diseases. A proper functional bone marrow niche is a fundamental requirement to guarantee an efficient HSC engraftment and hematological reconstitution. The co-infusion of MSCs has emerged as a feasible and safe therapeutic strategy to improve the HSCT outcome. Indeed, MSCs have been shown to support the engraftment of transplanted HSCs reducing the risk of graft failure by secreting soluble factors, and limiting the risk of GvHD thanks to their anti-inflammatory properties. However, several alterations of the stromal compartment have been described in malignant and nonmalignant diseases, which compromises MSC biological characteristics and hematopoietic supportive capacity. These findings have important implications for the clinical practice of HSCT. First, they suggest that BM stromal compartment associated defects may contribute to the reduced HSC engraftment leading to graft failure in some specific diseases, such as beta-thalassemia or other hematological disorders. Moreover, considering that the BM stroma plays a central role in the control of HSC homeostasis, the altered functionality of MSCs may negatively affect the hematopoietic dynamics of HSCPs following transplantation. Several cases also report that BM stroma contributes to hematological disease-relapse in donor cells after transplantation for malignant disorders by modifying the BM microenvironment in support of tumor cells. All these observations highlight the need to deeply study the BM microenvironment with the aim to restore the proper signaling to support HSPC engraftment and long-term hematopoiesis, improving the efficacy of HSC transplantation. Furthermore, considering the impaired function of MSCs in some specific diseased contexts, the co-administration of allogeneic, HD-derived MSCs is to be preferred in co-transplantation settings with the aim to ameliorate HSC engraftment in patients. 

## Figures and Tables

**Figure 1 jcm-09-00002-f001:**
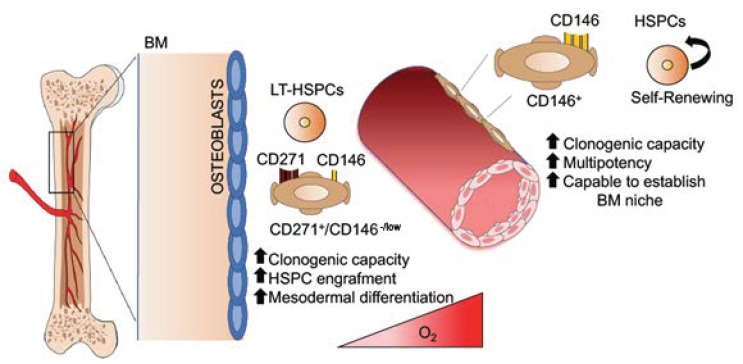
Schematic representation of a model describing mesenchymal stromal cells (MSCs) in the human bone marrow (BM) niche. In the human BM niche, different subtypes of MSCs interact with hematopoietic stem cells (HSCs) in the different perivascular regions and show specific functional characteristics. In particular, classification determinants (CD) 271 and CD271^+^/CD146^-/low^ MSCs are bone-lining MSCs associated with long-term (LT)-HSCs in low oxygen areas, whereas CD146^+^ and CD271^+^/CD146^+^ are located around the BM sinusoids in association with activated and fast-proliferating HSCs. Abbreviations: BM—bone marrow; MSC—mesenchymal stromal cells; HSC—hematopoietic stem cell; CD—classification determinant; LT—long-term; HSPC—hematopoietic stem and progenitor cell.

**Figure 2 jcm-09-00002-f002:**
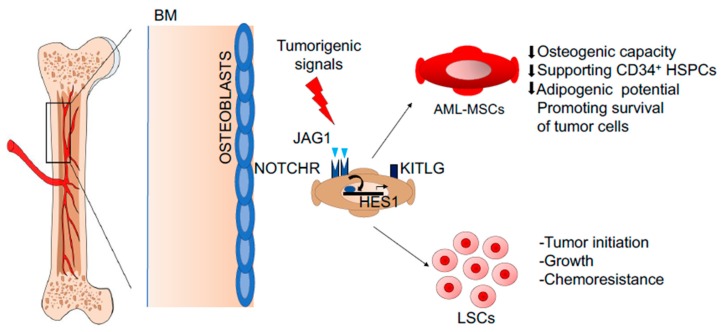
Representative image describing the role of the NOTCH/JAG1 pathway in AML-MSCs of a model describing MSCs in human BM niche. NOTCH/JAG1 signaling plays a fundamental role in the MSC-dependent control of tumor initiation, growth, and chemoresistance. NOTCH1, JAG1, and HES1 are overexpressed in AML-MSCs, promoting survival of tumor cells treated with chemotherapeutic agents. Abbreviations: NOTCH—Notch Homolog 1, Translocation-Associated; JAG1—of jagged canonical Notch ligand 1; AML-MSCs—acute myeloid leukemia derived mesenchymal stroma cells; MSC—mesenchymal stromal cells; BM—bone marrow; HES1—hairy and enhancer of split 1; KITLG—KIT ligand; LSCs—leukemic stem cells; HSPCs—hematopoietic stem and progenitor cells.

**Figure 3 jcm-09-00002-f003:**
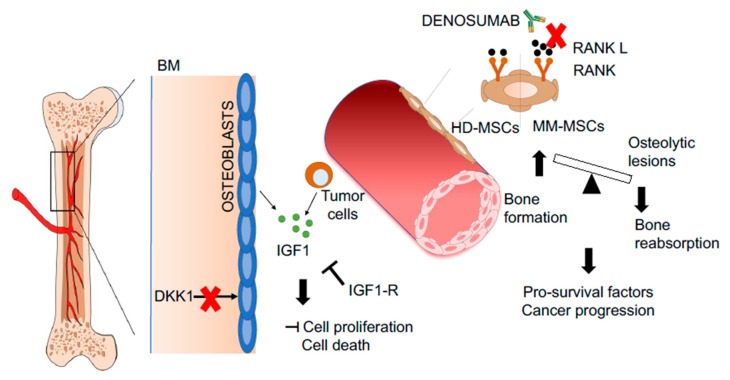
Representative pictures of novel therapeutic strategies targeting the malignant BM niche. Denosumab reduces osteolytic lesions by inhibiting the RANK–RANKL pathway (1). The mobilization of leukemic stem cells by plerixafor increase the efficacy of chemotherapy (2). Inhibition of DKK-1 prevents bone damage by inducing osteoblast differentiation (3). Inhibition of IGF1-R blocks the proliferation cascade activated by IGF1 released by osteoblast and tumor cells (4). Abbreviations: RANK—receptor activator of nuclear factor-kappa-Β ligand; RANKL—RANK ligand; DKK-1—Dickkopf-related protein 1; IGF1—insulin growth factor 1; IGF1-R—IGF1 receptor; BM—bone marrow; HD-MSCs—healthy donor-derived mesenchymal stromal cells; MM-MSCs—multiple myeloma-derived mesenchymal stromal cells.

**Figure 4 jcm-09-00002-f004:**
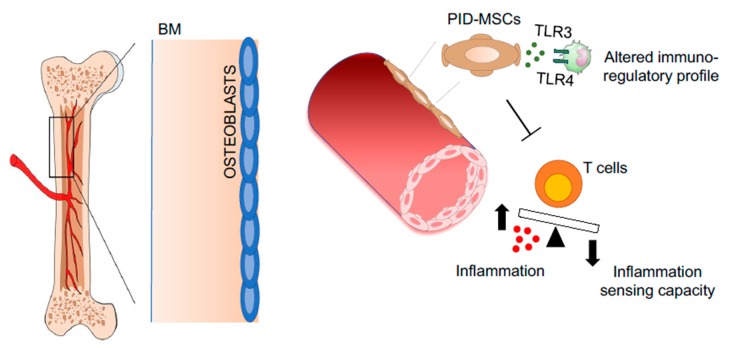
Representative pictures of pathological BM niches in PIDs. MSCs derived from primary immune-deficient (PIDs) patients, suffering from frequent infection and extensive inflammation, which compromises the capacity of MSCs to control T cell proliferation and alters their immunoregulatory profile. Abbreviations: BM—bone marrow; PID—primary immunodeficiencies; MSCs—mesenchymal stromal cells; TLR—Toll-like receptor.

**Figure 5 jcm-09-00002-f005:**
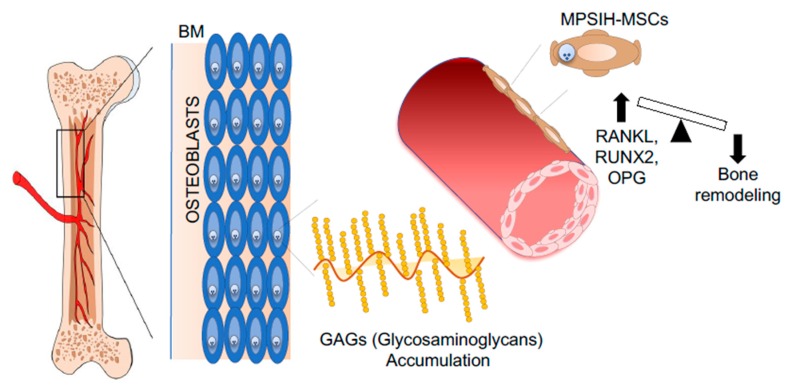
Representative pictures of pathological BM niches in MPSIH patients. MSCs derived from mucopolysaccharidosis type I- Hurler–syndrome (MPSIH) patients. The accumulation of Glycosaminoglycans (GAGs) causes multiorgan dysfunction, including bone defects. The upregulation of RANKL/RANK/OPG pathway causes unbalanced bone remodeling. Abbreviations: BM—bone marrow; MPSIH-MSCs—Mucopolysaccharidosis type I Hurler–derived mesenchymal stroma cells; RANK—receptor activator of nuclear factor-kappa-Β ligand; RANKL—RANK ligand; OPG—osteoprotegerin.

**Figure 6 jcm-09-00002-f006:**
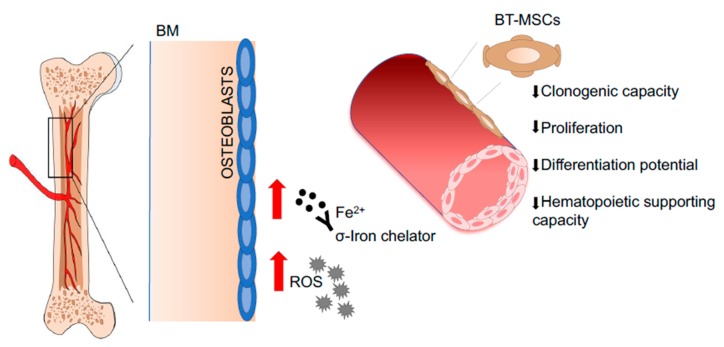
Representative pictures of pathological BM niches in BT patients. MSCs derived from Beta Thalassemia (BT) patients. Iron accumulation in the BM leads to increased levels of Radical Oxygen Species (ROS), which causes a pauperization of the primitive MSC pool and alters the MSC hematopoietic supportive function. Abbreviations: BM—bone marrow; BT-MSCs—beta-thalassemia derived mesenchymal stroma cells; Fe^2+^—iron, ferrous ion.

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
