# Peer review of "Bone Marrow-Derived Mesenchymal Stromal Cells: A Novel Target to Optimize Hematopoietic Stem Cell Transplantation Protocols in Hematological Malignancies and Rare Genetic Disorders"

_jcm, 2019, doi:10.3390/jcm9010002_

Round 1

Reviewer 1 Report

In this review entitled “Bone marrow-derived mesenchymal stromal cells: a novel target to optimize hematopoietic stem cell transplantation protocols in hematological malignancies and rare genetic disorders” by authors Stefania Crippa et al., the authors revised the role of MSCs in the orchestration of the BM niche and discussed the alterations in the mesenchymal compartment in specific disorders. This review further proposed novel therapeutic strategies targeting the malignant BM niches. This review manuscript will be of great interest to researchers working on MSCs and has the priority to be published in the Journal.

It would be good that the authors make some figures to illustrate the details of the findings.

Author Response

December 2nd 2019

RE:“Bone marrow-derived mesenchymal stromal cells: a novel target to optimize hematopoietic stem cell transplantation protocols in hematological malignancies and rare genetic disorders” by Crippa et al.

Dear Editor,

thank you very much for your letter concerning the review of the above reported manuscript.

            As requested, we have modified the manuscript and answered to the points raised by the Reviewers. We now submit the revised version that has been prepared with all the modifications reported in bold. We hope that all these changes can satisfy the Reviewers’ requests/criticism.

We herewith confirm that:

The authors have seen and approved the revised manuscript. No author has conflicts of interest. Neither this paper nor any similar paper has been submitted to any other printed or digital publication.

Looking forward to hearing from you at your earliest convenience, we thank you very much in advance for your kind attention.

Yours sincerely,

Maria Ester Bernardo

REVIEWER #1

Comments and Suggestions for Authors

In this review entitled “Bone marrow-derived mesenchymal stromal cells: a novel target to optimize hematopoietic stem cell transplantation protocols in hematological malignancies and rare genetic disorders” by authors Stefania Crippa et al., the authors revised the role of MSCs in the orchestration of the BM niche and discussed the alterations in the mesenchymal compartment in specific disorders. This review further proposed novel therapeutic strategies targeting the malignant BM niches. This review manuscript will be of great interest to researchers working on MSCs and has the priority to be published in the Journal.

It would be good that the authors make some figures to illustrate the details of the findings.

First of all we would like to thank the Reviewer for having appreciated our manuscript and for this useful suggestion. As proposed, we have expanded the number of figures up to 6 in order to illustrate some of the fundamental interactions within the healthy and “diseased” BM niche. We hope that this gives a better graphical explanation of the proposed mechanisms.

Moreover, we have carefully revised English throughout the manuscript in order to make it more fluent and clear.

Reviewer 2 Report

Bone marrow-derived Mesenchymal Stromal Cells: a novel target to optimize hematopoietic stem cell transplantation protocols in hematological malignancies and rare genetic disorders

Stefania Crippa, Ludovica Santi, Roberto Bosotti, Giulia Porro, Maria Ester Bernardo

Feedback for JCM

Specific comments

Page 3 Line 78-79 – “and capable to generate a BM niche supporting hematopoietic activity when transplanted heterotopically..” replace “capable to” with “can” e.g. “and can generate a BM niche..”

Page 3 Line 82-82 – “CD271+ MSCs show an enhanced clonogenic capacity, ability to differentiate into osteoblasts, chondrocytes and adipocytes and express extracellular matrix and cell adhesion genes (Jones et al., 2002) (Kuci et al., 2010) (Ganguly et al., 2019)”

The differentiation pattern is standard and the outlined prerequisite pattern for MSCs differentiation by the ISCT. This is not telling us anything extra about the CD271+ cells over. Suggest remove or rewrite to exemplify differences.

Page 3 Line 88 – “Increasing evidences support” change to “Increasing evidence supports”

Page 4 Line 94 Good idea to refresh the relative natural abundance, reads well.

Page 4 Line 96 – “Increasing evidences suggest” change to “Increasing evidence suggests”

Page 4 Line 99 – “It has been shown that MSC cultures undergo clonal selection during the expansion phase, and selected clones possess different capabilities (Selich et al., 2019).” Excellent summary and very important point.

Page 4 112-116 – “MSCs have been employed as tools for tissue engineering and regenerative medicine (Rohban and Pieber, 2017) (Steinert et al., 2012) (Martin et al., 2019, as well as cell-therapy approach to counteract inflammation in immune-mediated and inflammatory diseases thanks to their capacity of sensing the inflammatory state and suppressing immune responses {Nauta, 2007 #150) (Le Blanc and Davies, 2015) (Bernardo and Fibbe, 2013). This section needs significant work, not least the references – refer general comments and ensure references are numerical.

Page 4 Line 116 – “Moreover, several pre-clinical and clinical data highlighted” change data to studies

Page 4 Line 122 -  “pathological contests” change contests maybe to conditions

Page 6 Line 166 – “Despite the study of human BM niche is challenged by their difficult accessibility” need to re word suggest “.. of the human BM niche it is challenged..”

Page 6 Line 176 – “However, the majority of CD271 175 + MSCs do not co-express CD90 and CD73 and this marker is not universally expressed in different MSCs populations, suggesting that CD271 is not sufficient to isolate MSCs from various tissues”. Needs a little more expansion and contextualisation regarding defining characteristics of MSC’s and ISCT definitions

Page 6 Line 184 – “CD271+ and CD271+ /CD146-/low MSCs have been reported to be bone-lining cells associated with long-term (LT)-HSPCs in low oxygen areas, whereas CD146+ and CD271+/CD146+ are located around BM sinusoids in association with proliferating HSPCs. In both regions”. Requires illustrative figure. I understand this is not fully defined but this is a review and thus a summary of the current status quo. You can always make a qualifying statement such as “evidence to date suggests” etc.

Page 7 Line 209 “several differences exist according to MSCs’ tissue sources”. Change exist to existing.

Page 7 Lines 218 onwards. GvHD clinical trial information needs fuller clarification. Studies need to be exemplified as allogenic or autologous at all juncture.

Page 7-8 Lines 222-224. The way this is written it seems you are suggesting that on no occasion are any adverse events reported in any clinical trials with MSCs (either allogeneic or autologous). Then the paragraph moves into timings of administration. This ection needs to be better written and have clearer definition.

Page 8 Line 229. GvHD please stay consistent with abbreviations.

Page 8 Line 231. Spelling “immuneregulatory”

Page 8 Line 255. Needs a space before reference but this should be sorted when all the references are put in the correct format.

Page 9 Line 260. “. In these cases, MSCs are thought to promote regeneration by secreting paracrine factors (Gunawardena et al., 2019), despite there is no clear explanation of their mechanisms of action in these settings.” Reword sentence for clarity. No explanation or no consensus available as to MoA?

Page 9 Line 226. Contest should be context

Page 9 Line 274 onwards. Good concluding statement but please add in some more references such as Samsonraj et al., and Bahsoun et al., when talking about MSC manufacture for use in clinic.

Page 9 Line 285 change the word dispense, maybe for expense. Will lend clarity to the English.

Page 9 to 10. Some strong summaries well written.

Page 10. A good depth of detail, very informative. Just be careful not to diminish such strong writing towards the end. Refer Page 11 Lines 329-332 this could be better written to provide as much impact as the preceding section.

Page 11 Line 340. Suggest result (singular) rather than results.

Page 12 Line 377 – “MSC functional characteristic” change to characteristics.

Page 12 Line 390 – “Denosumab is capable to ameliorate bone disease”. English is confusing here. Try “is capable of ameliorating bone disease”

Page 14 Line 249 onwards. Very enjoyable.

Page 14 Line 456-459 – “Despite PID-MSCs do not show any alterations in terms of proliferation, differentiation, expression of MSC surface marker, the immunomodulatory capacity of PID-MSCs was altered compared to age-matched controls (Ingo et al., 2016) (Starc et al., 2017). This needs rewriting at present it makes little sense. You could try “ Despite the fact that PID-MSCs do not display…”

Page 15 Line 476 – “Despite MPSIH-MSCs are able to differentiate into osteoblasts, an upregulation of RANKL/RAN/OPG has been found in patient-derived MSCs compared to controls”. Please rework to lend clarity to what you are saying/stating.

Page 15 Line 489 – 494. Two sentences running together which seem to be making the same statement. Please merge and make concise.

Page 16 Line 498 – “Treatment of BT-MSCs with an iron chelator partially restores their functionality”. Seems a throw away statement in the wider paragraph. Please either remove or expand.

Figure 2 needs to be a clearer image of higher quality especially the text.

General comments

Should references be numbered and merged into a single set of parentheses for example Page 4 Line 109 (Zhou et al., 2017) (Li et al., 2007) (Lv et al., 2015) would become (13-15 – or whichever number they are). All references need to be in the journals correct format and numerical, please alter.

On occasion the English is confusing and needs correcting (see specific comments). Please read through carefully on completion to ensure that this does not hinder a good review.

Abbreviations. Please adhere to expansion on fits usage and the Journals abbreviations policies. Also ensure that you stay consistent within your abbreviations.

Consider adding “narrative review” to title.

Figures need to be cited clearly in text (Figures 1 and 2).

Consider expanding the number of figures in the review overall. You write some strong details about signalling pathways and the interplay between BM niche and cellular content consider if you feel you could add more illustrative detail.

Author Response

December 2nd 2019

RE:“Bone marrow-derived mesenchymal stromal cells: a novel target to optimize hematopoietic stem cell transplantation protocols in hematological malignancies and rare genetic disorders” by Crippa et al.

Dear Editor,

thank you very much for your letter concerning the review of the above reported manuscript.

            As requested, we have modified the manuscript and answered to the points raised by the Reviewers. We now submit the revised version that has been prepared with all the modifications reported in bold. We hope that all these changes can satisfy the Reviewers’ requests/criticism.

We herewith confirm that:

The authors have seen and approved the revised manuscript. No author has conflicts of interest. Neither this paper nor any similar paper has been submitted to any other printed or digital publication.

Looking forward to hearing from you at your earliest convenience, we thank you very much in advance for your kind attention.

Yours sincerely,

Maria Ester Bernardo

REVIEWER #2

Specific comments

- Page 3 Line 78-79 – “and capable to generate a BM niche supporting hematopoietic activity when transplanted heterotopically..” replace “capable to” with “can” e.g. “and can generate a BM niche..”

We changed the text according to reviewer’s suggestion

- Page 3 Line 82-82 – “CD271+ MSCs show an enhanced clonogenic capacity, ability to differentiate into osteoblasts, chondrocytes and adipocytes and express extracellular matrix and cell adhesion genes (Jones et al., 2002) (Kuci et al., 2010) (Ganguly et al., 2019)”

The differentiation pattern is standard and the outlined prerequisite pattern for MSCs differentiation by the ISCT. This is not telling us anything extra about the CD271+ cells over. Suggest remove or rewrite to exemplify differences.

We thank the reviewer for this suggestion. In the revised version of the manuscript we rewrote the text to clearly explain the characteristics of human CD271+ MSCs:CD271+ MSCs show an enhanced clonogenic and differentiation capacity, and express higher levels of extracellular matrix and cell adhesion genes compared to bulk MSCs (Jones et al., 2002) (Kuci et al., 2010) (Ganguly et al., 2019).

- Page 3 Line 88 – “Increasing evidences support” change to “Increasing evidence supports”

We changed the text according to reviewer’s suggestion.

- Page 4 Line 94 Good idea to refresh the relative natural abundance, reads well.

We thank the Reviewer for his/her positive comment

- Page 4 Line 96 – “Increasing evidences suggest” change to “Increasing evidence suggests”

We changed the text according to reviewer’s suggestion.

- Page 4 Line 99 – “It has been shown that MSC cultures undergo clonal selection during the expansion phase, and selected clones possess different capabilities (Selich et al., 2019).” Excellent summary and very important point.

We thank the reviewer for the positive comment.

- Page 4 112-116 – “MSCs have been employed as tools for tissue engineering and regenerative medicine (Rohban and Pieber, 2017) (Steinert et al., 2012) (Martin et al., 2019, as well as cell-therapy approach to counteract inflammation in immune-mediated and inflammatory diseases thanks to their capacity of sensing the inflammatory state and suppressing immune responses {Nauta, 2007 #150) (Le Blanc and Davies, 2015) (Bernardo and Fibbe, 2013). This section needs significant work, not least the references – refer general comments and ensure references are numerical.

We thank the reviewer for this suggestion. In the revised version of the manuscript we have briefly described the immunomodulatory and anti-inflammatory properties of MSCs, including references of instrumental works demonstrating the capacity of MSCs to sense the surrounding environment and control inflammation. We did not go into details because the manuscript aims at describing the effect of alterations in the BM stroma on the progression of hematological disorders and genetic disease. Moreover, further explanation of the immuneregulatory abilities of MSCs is reported on page 7 of the manuscript (lines 257-263).

- Page 4 Line 116 – “Moreover, several pre-clinical and clinical data highlighted” change data to studies

We changed the text according to reviewer’s suggestion.

- Page 4 Line 122 -  “pathological contests” change contests maybe to conditions

We changed the text according to reviewer’s suggestion.

- Page 6 Line 166 – “Despite the study of human BM niche is challenged by their difficult accessibility” need to re word suggest “.. of the human BM niche it is challenged..”

We changed the text according to reviewer’s suggestion.

- Page 6 Line 176 – “However, the majority of CD271+ MSCs do not co-express CD90 and CD73 and this marker is not universally expressed in different MSCs populations, suggesting that CD271 is not sufficient to isolate MSCs from various tissues”. Needs a little more expansion and contextualisation regarding defining characteristics of MSC’s and ISCT definitions

We thank the reviewer for this suggestion. In the revised version of the manuscript we changed the text, as follows: “However, the majority of CD271+ MSCs do not co-express CD90 and CD73, two of the markers included in the minimal criteria used to define ex-vivo expanded MSCs, according to the International Society for Cellular Therapy position paper (Dominici et al., 2006). For this reason, CD271+ expression is not an adequate marker to prospectively isolate bona-fide MSCs from BM samples, although it is not clear whether the expression of CD90, CD73, and CD105 represents an in vitro artifact. Preliminary data obtained in our laboratory demonstrate the existence of a rare population of cells positive for CD90, CD73, CD105 in the CD34 negative fraction of BM aspirates (Crippa et al, in preparation). Moreover, CD271 shows different expression levels among various MSC subsets and it is not universally expressed in MSCs derived from different tissues, suggesting that this marker is not sufficient to prospectively isolate MSCs (Quirici et al., 2002) (Lv et al., 2014)”.

- Page 6 Line 184 – “CD271+ and CD271+ /CD146-/low MSCs have been reported to be bone-lining cells associated with long-term (LT)-HSPCs in low oxygen areas, whereas CD146+ and CD271+/CD146+ are located around BM sinusoids in association with proliferating HSPCs. In both regions”. Requires illustrative figure. I understand this is not fully defined but this is a review and thus a summary of the current status quo. You can always make a qualifying statement such as “evidence to date suggests” etc.

We thank the reviewer for this meaningful suggestion. In the revised version of the manuscript we have included Figure 1 to describe MSC subpopulations in the human BM niche and we have added the statement “evidence to date suggests” to strengthen our phrasing.

- Page 7 Line 209 “several differences exist according to MSCs’ tissue sources”. Change exist to existing.

We changed the text according to reviewer’s suggestion.

- Page 7 Lines 218 onwards. GvHD clinical trial information needs fuller clarification. Studies need to be exemplified as allogenic or autologous at all juncture.

We thank the Reviewer for highlighting this point. In the revised version of the manuscript we have provided more details on the clinical trials based on MSC treatment for acute GvHD, including information on the clinical context (allogeneic HSCT) and level of matching of the MSC donor with the treated patients.

- Page 7-8 Lines 222-224. The way this is written it seems you are suggesting that on no occasion are any adverse events reported in any clinical trials with MSCs (either allogeneic or autologous). Then the paragraph moves into timings of administration. This section needs to be better written and have clearer definition.

We thank again the Reviewer for making an important point. In the revised version of the manuscript we have blunted the sentence on MSC-related adverse events.

- Page 8 Line 229. GvHD please stay consistent with abbreviations.

In the revised version we have checked thoroughly the abbreviation for GvHD.

- Page 8 Line 231. Spelling “immuneregulatory”

We corrected the misspelled words.

- Page 8 Line 255. Needs a space before reference but this should be sorted when all the references are put in the correct format.

We added a space before the reference.

- Page 9 Line 260. “. In these cases, MSCs are thought to promote regeneration by secreting paracrine factors (Gunawardena et al., 2019), despite there is no clear explanation of their mechanisms of action in these settings.” Reword sentence for clarity. No explanation or no consensus available as to MoA?

We thank the reviewer for this useful suggestion. We have reworded the sentence for better clarity.

- Page 9 Line 226. Contest should be context

We changed the text according to reviewer’s suggestion.

- Page 9 Line 274 onwards. Good concluding statement but please add in some more references such as Samsonraj et al., and Bahsoun et al., when talking about MSC manufacture for use in clinic.

We thank the reviewer for this useful suggestion. In the revised version of the manuscript we included the following references:

Samsonray, R.M. et al. Concise Review: Multifaceted Characterization of Human Mesenchymal Stem Cells for Use in Regenerative Medicine. Stem Cells Transl Med. 2017

Bahsoun, S. et al. The Role of Dissolved Oxygen Levels on Human Mesenchymal Stem Cell Culture Success, Regulatory Compliance, and Therapeutic Potential. Stem Cells Dev. 2018

- Page 9 Line 285 change the word dispense, maybe for expense. Will lend clarity to the English.

We changed the text according to reviewer’s suggestion.

- Page 9 to 10. Some strong summaries well written.

We thank the reviewer for the positive comment.

- Page 10. A good depth of detail, very informative. Just be careful not to diminish such strong writing towards the end. Refer Page 11 Lines 329-332 this could be better written to provide as much impact as the preceding section.

We changed the text according to reviewer’s suggestion. In the revised version of the manuscript we better clarified the impact of the alterations affecting the immunoregulatory properties of neoplastic BM stromal niche, as follows: “Moreover, several studies demonstrated an altered immunomodulatory function of MDS-MSCs, affecting tumor immune surveillance (Fracchiolla et al., 2017). Several works demonstrated that MSCs from neoplastic niche are educated to polarize macrophage toward an M2, anti-inflammatory, phenotype and they are responsible for increase Treg through the secretion of indoleamine 2,3-dioxygenase (IDO). In addition, MDS-MSCs have been shown to inhibit dendritic cell maturation and alter their functions, including including endocytosis, IL-12 secretion, their ability to inhibit T cell proliferation. (Zhao et al., 2012) (Wang et al., 2013)(Ciciarello et al. 2019). In this light, BM stroma contributes in the creation of a protective and immune-tolerant microenvironment capable to support the survival of leukemic cells and affect the response to therapies.

- Page11 Line 340. Suggest result (singular) rather than results.

We changed the text according to reviewer’s suggestion.

- Page 12 Line 377 – “MSC functional characteristic” change to characteristics.

We changed the text according to reviewer’s suggestion.

- Page 12 Line 390 – “Denosumab is capable to ameliorate bone disease”. English is confusing here. Try “is capable of ameliorating bone disease”

We changed the text according to reviewer’s suggestion.

- Page 14 Line 249 onwards. Very enjoyable.

We thank the reviewer for the positive comment.

- Page 14 Line 456-459 – “Despite PID-MSCs do not show any alterations in terms of proliferation, differentiation, expression of MSC surface marker, the immunomodulatory capacity of PID-MSCs was altered compared to age-matched controls (Ingo et al., 2016) (Starc et al., 2017). This needs rewriting at present it makes little sense. You could try “ Despite the fact that PID-MSCs do not display…”

We changed the text according to reviewer’s suggestion.

- Page 15 Line 476 – “Despite MPSIH-MSCs are able to differentiate into osteoblasts, an upregulation of RANKL/RAN/OPG has been found in patient-derived MSCs compared to controls”. Please rework to lend clarity to what you are saying/stating.

We changed the text in order to make it more readable and clear.

Page 15 Line 489 – 494. Two sentences running together which seem to be making the same statement. Please merge and make concise.

We changed the text according to reviewer’s suggestion.

Page 16 Line 498 – “Treatment of BT-MSCs with an iron chelator partially restores their functionality”. Seems a throw away statement in the wider paragraph. Please either remove or expand.

In the revised version of the manuscript we removed the indicated statement.

Figure 2 needs to be a clearer image of higher quality especially the text.

In the revised version of the manuscript we provided a higher quality image.

General comments

Should references be numbered and merged into a single set of parentheses for example Page 4 Line 109 (Zhou et al., 2017) (Li et al., 2007) (Lv et al., 2015) would become (13-15 – or whichever number they are). All references need to be in the journals correct format and numerical, please alter.

References have been adjusted according to the Journal format.

On occasion the English is confusing and needs correcting (see specific comments). Please read through carefully on completion to ensure that this does not hinder a good review.

We have corrected the English as suggested by the Reviewer and we have carefully read thoroughly the manuscript to render it clearer and fluent.

Abbreviations. Please adhere to expansion on fits usage and the Journals abbreviations policies. Also ensure that you stay consistent within your abbreviations.

We have verified abbreviations in order to be consistent.

Consider adding “narrative review” to title.

As suggested we have added “narrative” to the word “review” on the paper front paper.

Figures need to be cited clearly in text (Figures 1 and 2).

All the figures (Fig. 1-6) have now been cited properly in the text.

Consider expanding the number of figures in the review overall. You write some strong details about signalling pathways and the interplay between BM niche and cellular content consider if you feel you could add more illustrative detail.

As proposed by the Reviewer, we have expanded the number of figures up to 6 in order illustrate some of the fundamental interactions within the healthy and “diseased” BM niche. We hope that this gives a better graphical explanation of the proposed mechanisms.

Round 2

Reviewer 2 Report

Bone marrow-derived Mesenchymal Stromal Cells: a novel target to optimize hematopoietic stem cell transplantation protocols in hematological malignancies and rare genetic disorders (644670)

Stefania Crippa, Ludovica Santi, Roberto Bosotti, Giulia Porro, Maria Ester Bernardo

Feedback for JCM

General comments

This is a very good review of a very important but not yet fully recognised clinical target. There are some high level details about signalling pathways and the interplay between BM niche and cellular content shown in more illustrative detail and the English is significantly improved. This review manuscript will be of great value to researchers working on MSCs.

Author Response

December 11st 2019

RE:“Bone marrow-derived mesenchymal stromal cells: a novel target to optimize hematopoietic stem cell transplantation protocols in hematological malignancies and rare genetic disorders” by Crippa et al.

Dear Editor,

thank you very much for your second letter concerning the review of the above reported manuscript.

            As requested, we have modified the manuscript and answered to the points raised by the Reviewers. In particular we have defined all the abbreviations and we have added pieces of information in the text where requested. A paragraph with the conclusions from the reported findings has been also added.

We now submit the revised version that has been prepared with the main modifications higlighted in yellow. We hope that all these changes can satisfy the Reviewers’ requests/criticism.

We herewith confirm that:

The authors have seen and approved the revised manuscript. No author has conflicts of interest. Neither this paper nor any similar paper has been submitted to any other printed or digital publication.

Looking forward to hearing from you at your earliest convenience, we thank you very much in advance for your kind attention.

Yours sincerely,

Maria Ester Bernardo
